# De Novo Assembly and Annotation of 11 Diverse Shrub Willow (*Salix*) Genomes Reveals Novel Gene Organization in Sex-Linked Regions

**DOI:** 10.3390/ijms24032904

**Published:** 2023-02-02

**Authors:** Brennan Hyden, Kai Feng, Timothy B. Yates, Sara Jawdy, Chelsea Cereghino, Lawrence B. Smart, Wellington Muchero

**Affiliations:** 1Horticulture Section, School of Integrative Plant Science, Cornell University, Geneva, NY 14456, USA; 2Biosciences Division, Oak Ridge National Laboratory, Oak Ridge, TN 37830, USA

**Keywords:** *Salix*, shrub willow, genome assembly, sex determination

## Abstract

Poplar and willow species in the Salicaceae are dioecious, yet have been shown to use different sex determination systems located on different chromosomes. Willows in the subgenus *Vetrix* are interesting for comparative studies of sex determination systems, yet genomic resources for these species are still quite limited. Only a few annotated reference genome assemblies are available, despite many species in use in breeding programs. Here we present de novo assemblies and annotations of 11 shrub willow genomes from six species. Copy number variation of candidate sex determination genes within each genome was characterized and revealed remarkable differences in putative master regulator gene duplication and deletion. We also analyzed copy number and expression of candidate genes involved in floral secondary metabolism, and identified substantial variation across genotypes, which can be used for parental selection in breeding programs. Lastly, we report on a genotype that produces only female descendants and identified gene presence/absence variation in the mitochondrial genome that may be responsible for this unusual inheritance.

## 1. Introduction

The genus *Salix* and the Salicaceae family are of growing scientific interest for their use as model systems to understand sex determination and sex chromosome dynamics. The Salicaceae family is almost entirely dioecious, and contains approximately 30 species of *Populus* and over 300 species of *Salix* [1], yet both the location of the sex determination system (SDR) as well as the sex inheritance mechanism (ZW vs. XY) differ across clades within this family. In the subgenus *Vetrix*, the sex determination region has been localized to Chr15, with a ZW system of inheritance [2,3], while Chr15XY has been identified in *S. triandra* from the subgenus *Salix* [4,5] and Chr07 XY in subgenus *Protitea* (*S. nigra* [6], *S. chaenomeloides* [7], and *S. dunnii* [8]) and subgenus *Pleuradenia* (*S. arbutifolia*) [4,9]. In *Populus*, Chr19 XY, Chr19 ZW, and Chr14 XY sex determination systems have all been reported [10]. The precise genes responsible for sex determination in Salicaceae are still being studied but are thought to involve a presence/absence of expression of the type-C cytokinin response regulator *ARR17* in *Populus* [11] and *Salix* subgenus *Protitea* [7], in which *ARR17* acts as a single gene controlling sex, simultaneously promoting female floral development and suppressing male floral development. Willows in the subgenus *Vetrix*, on the other hand, may possess a two-gene system of sex determination where two genes on the W chromosome are necessary to determine sex: one to promote female floral development and another to suppress male development. Dosage levels of *ARR17* in combination with *GATA15* have been suggested as a possible two-gene mechanism of sex determination in the shrub willow *S. purpurea*, based on expression and resequencing evidence from a set of monoecious families in this species [12]. *AGO4*, *DRB1*, and three hypothetical proteins have also been proposed as potential master regulators of sex in *S. purpurea* [13]. 

Shrub willows in the subgenus *Vetrix* are a dioecious crop grown widely across the northern hemisphere for a variety of horticultural uses, including for bioenergy, as ornamentals, and for ecological restoration purposes [1]. Commonly cultivated shrub willow species include European natives *S. purpurea* and *S. viminalis*, the Chinese species *S. suchowensis*, Japanese natives *S. integra* and *S. udensis*, and *S. koriyanagi* from Korea [1,14,15]. Together, these six aforementioned species represent a broad range of genetic diversity across *Vetrix* [2,14]. Due to both the dynamic nature of the SDR within this genus and the unique mechanism of sex determination in *S. purpurea* [12,13], there is an interest in comparing the gene content of the sex determination regions across species in *Vetrix*, in order confirm the two-gene model from *S. purpurea* and to identify any additional shifts in sex determination genes during the evolution of this clade.

*Salix* are both wind and insect pollinated and as such, catkins produce a suite of secondary metabolites to attract pollinators [16,17,18]. Previous studies that characterized terpenoid and flavonoid profiles in *Salix* catkins have shown substantial differential expression of these compounds based on sex, which influences pollinator attraction [16,19]. Secondary metabolites also play a known role in defense against herbivory across plant species [20,21]. QTL mapping of floral terpenoid, flavonoid, and phenolic glucoside production and identification of candidate genes have been conducted in *S. purpurea,* and candidate genes for many specific compounds have been identified [19]. However, as of yet, there has been little effort to compare these candidate genes between related species. Characterizing the presence, copy number, and expression of secondary metabolite genes across *Salix* species is therefore useful for understanding biological differences in floral secondary metabolite production, and their effects on pollinator attraction and herbivory. 

Genomic resources for the genus *Salix* are still under development, with the shrub willows being the most well-studied group with several assembled genomes and recent advances in QTL mapping of various traits, including yield, insect resistance, and rust susceptibility [22,23]. Within the subgenus *Vetrix*, reference genomes are currently available for a female *S. viminalis* [24], a female *S. suchowensis* [25], a male *S. purpurea* (‘Fish Creek’), a female *S. purpurea* (94006) [26], and a monoecious *S. purpurea* [12], the latter two of which have fully assembled Chr15Z and Chr15W sex chromosomes. Here we present de novo assembly and annotation of 11 *Salix* genomes across six shrub willow species, including three newly sequenced and assembled species. Among these 11 genomes is a reassembly of 94006, a *S. purpurea* female that was used for the Phytozome v5.1 reference genome (https://phytozome-next.jgi.doe.gov/info/Spurpurea_v5_1, last accessed 2 December 2022) and is the mother of the male ‘Fish Creek’ used for the Phytozome v3.1 reference (https://phytozome-next.jgi.doe.gov/info/SpurpureaFishCreek_v3_1, last accessed 2 December 2022), both produced by the US Department of Energy Joint Genome Institute (JGI) [26,27]. A male *S. purpurea*, (94001, the father of ‘Fish Creek’), two female (P294, P295) and one male (P63) *S. suchowensis*, one female *S. integra* (P336), one male (04-FF-016) and one female (SH3) *S. koriyanagi*, one female (07-MBG-5027) and one male (‘Jorr’) *S. viminalis*, and a male *S. udensis* (04-BN-051) were also sequenced. These particular genotypes were utilized since previous research has reported Chr15 ZW sex inheritance for each of these genotypes and their SDR boundaries have been delineated [2]. Furthermore, F_1_ crosses to *S. purpurea* 94001 and 94006 have been previously generated for each genotype, with their genetic linkage maps, phenotypic analysis, and QTL mapping results published [2].

For each assembly and annotation, gene content across the Chr15W SDR regions was characterized. Notably, nearly all previously identified candidate sex determination genes are missing from *S. koriyanagi, S. viminalis*, and *S. udensis*, which suggests a unique sex determination mechanism in these species that may not involve *ARR17,* as shown in *Populus* and *S. purpurea* [12,28]. We report the expression and copy number variation of multiple secondary metabolite genes, including previously identified candidates for sex-dimorphic floral volatile and phenolic glycoside compound production [19]. Finally, we present data that support an all-female inheritance in the *Salix integra* P336 descendants and identify a missing mitochondrial *RPL10* gene as a candidate mechanism for this inheritance. 

## 2. Results

### 2.1. Assembly and Annotation

Oxford Nanopore read length and quality distributions for each assembly are shown in Appendix A, respectively. Mean genome coverage ranged from 45× to 103×. Contig N50 values ranged from 300.36 Kb in 04-FF-016, to 804.25 kb in P336. Assembly lengths were relatively consistent within each species. *S. suchowensis* had the largest genome size, with a mean of 375 Mb, while the mean size of the *S. viminalis* genome was only 288 Mb. All assemblies had a Eudicot core gene BUSCO score above 95%. Assembly statistics are shown in Table 1. 

Annotation BUSCO scores ranged from 77.9% in P336 (*S. integra*) to 92.9% in ‘Jorr’ (*S. viminalis*). The mean number of annotated genes across all genomes was 32,166, while the mean number of annotated transcripts was 40,679. The estimated number of missing genes, relative to 94006 v5.1, ranged from 3706 in the 94006 reassembly (*S. purpurea*) to 4973 in SH3 (*S. koriyanagi*). The total gene number in genome-specific orthogroups ranged from 331 in ‘Jorr’ (*S. viminalis*) to 1026 in 94006 (*S. purpurea*). Annotation statistics are shown in Table 2. 

Comparative genomics analysis with Orthofinder assigned 391,057 out of 407,955 transcripts across all 11 annotations to 49,209 orthogroups (95.2% of transcripts). A total of 2769 orthogroups were genome-specific and accounted for 1.7% of all transcripts. Orthogroup assignment had a G50 of 11 (i.e., 50% of genes were assigned to an orthogroup of 11 or larger) and an O50 of 11,958 among assigned genes (i.e., 11,958 orthogroups accounted for 50% of genes). A total of 10,799 orthogroups had genes from all 11 genomes represented, 3401 of which were single-copy orthogroups. Phylogenetic analysis of the annotated gene sets using Orthofinder consistently grouped genomes of the same species together (Appendix A).

### 2.2. Sex Determination Gene Analysis

Copy numbers of each candidate sex determination gene in the Chr15W locus varied among genotypes, as did the Chr15Z exon 1 and Chr19 full-length copies of *ARR17* (Table 3). Candidate genes were present in expected numbers in *S. purpurea* 94006, consistent with the JGI *S. purpurea* 94006 v5.1 reference genome, with the exception of one fewer *ARR17* copy as well as *GATA15* initially assembled on Chr17 instead of Chr15W, which are likely due to errors during assembly or scaffolding. *S. suchowensis* P294, *S. suchowensis* P295, and *S. integra* P336 have two Chr15 *ARR17* copies and one Chr15 *AGO4* copy. Most candidate sex determination genes, including *ARR17*, *AGO4*, *GATA15*, were missing from *S. koriyanagi* SH3 and *S. viminalis* 07-MBG-5027 (Table 3). All Chr15 gene copies were located within the boundaries of the sex determination regions of each genotype, as previously described by Wilkerson et al. [2]. 

### 2.3. Secondary Metabolism Gene Analysis

BLASTN analysis of secondary metabolism genes revealed variation in copy number between genomes for most genes (Appendix A). The total combined expression of secondary metabolism genes across all eight tissue types also showed substantial variation between genotypes (Appendix A). 

### 2.4. P336 Crosses and Progeny

All progeny in the eight families generated with P336 (*S. integra*) as the female parent, including F_1_ progeny and second-generation progeny, were female, with over 75% of plants flowering in each family at the time of data collection (Table 4). 

## 3. Discussion

### 3.1. Assemblies and Annotations

The high quality of the assemblies (BUSCO > 95%) as well as the large number of genes in the annotations, represents an advancement in *Salix* genomic resources, including comprehensive comparative genome analysis across the shrub willows. Across all 11 annotations, there are several thousand gene models from *S. purpurea* 94006 v5.1 that are missing, which is reflected by relatively low BUSCO scores of less than 90% in most annotations (Table 2). The RNA-Seq data used to perform the annotations did not contain any floral tissue, nor any tissue from drought, disease, or insect stressed plants, which can explain the missing gene models, as genes from these biological conditions were not expressed in our dataset and therefore were not annotated. 

### 3.2. Sex Determination Genes and SDR Assembly

The reported assemblies each include one haplotype of Chr15 per genome: Chr15Z in the male assemblies and Chr15W in the females. Together, these include separate fully assembled 15Z and 15W chromosomes for *S. purpurea, S. suchowensis, S. koriyanagi*, and *S. viminalis*, Chr15W for *S. integra*, and Chr15Z for *S. udensis*. This is the first report of a fully assembled Chr15Z for both *S. suchowensis* and *S. viminalis* [24,25]. The Chr15 assemblies across the 11 genomes show substantial differences in structural arrangement (Appendix A–K). These structural differences may be due in part to errors in assembly rather than true structural variations between genotypes, particularly since the order of sequences in the reassembly of *S. purpurea* 94006 Chr15W differs from the JGI *S. purpurea* 94006 v5.1 assembly. Sex determination regions are notoriously difficult to assemble due to highly repetitive regions resulting from a lack of recombination, and such differences in arrangement of contigs into the final scaffolded sex chromosomes are not unexpected [29]. Nevertheless, despite structural variation, the Chr15 appears to be fully intact across every assembly. 

BLASTN results for candidate sex determination genes reveal substantial variation in gene content within the Chr15W SDR between genomes (Table 3). In *S. purpurea* 94006, the sex determination gene content closely matches the JGI *S. purpurea* 94006 v5.1 reference genome. Only three copies of *ARR17* were identified on Chr15 instead of four, five copies of *DRB1* instead of two, and *GATA15* was initially located on Chr17 instead of Chr15; however, these differences in gene copy number between assemblies could be the result of errors in assembly within the Chr15 in either reference. In the case of the missing fourth *ARR17*, this gene is located within a series of four palindromic repeats, and, due to their repetitive nature, the fourth arm could have been lost during haplotig purging. When searching the purged contigs, an additional *ARR17* was identified, which is likely this fourth Chr15 copy. In the case of *GATA15*, the Chr15W copy was in a 482 kb region that was originally assembled on Chr17. This was determined to be an error in the assembly, as no Chr17 *GATA15* was present in any other *S. purpurea* genome assembly, including the JGI 94006 v1.0 and v5.1 assemblies, the JGI ‘Fish Creek’ v3.1 assembly [26], the 94003 assembly [12], or our 94001 assembly. A dotplot alignment of HiC_scaffold_7 (Chr17) from our reassembly of 94006 against the JGI 94006 v5.1 reference shows this 482 kb region on HiC_scaffold_7 aligns to Chr15W (Appendix A). Linkage map markers for the *S. purpurea* 94006 genotype were obtained from Wilkerson et al. (2022) and include one marker, S15_7998352, which is located in the misassembled region [2]. In a BLASTN analysis, the flanking regions of this marker align to HiC_scaffold_7 (Chr17), while the nearest markers on the 94006 linkage map, which are tightly linked, align to HiC_scaffold_3 (Chr15), confirming that this region, including *GATA15*, is indeed a Chr15W region misassembled onto Chr17. In light of this finding, the 482 kb misassembled region on HiC_scaffold_7 was manually moved to HiC_scaffold_3. 

The *ARR17* and *GATA15* genes were absent from Chr15 in the males and present in the females of *S. purpurea* and *S. suchowensis*, consistent with the two-gene sex determination mechanism proposed by Hyden et al. [12,13] and suggesting a common sex determination mechanism between these two species. *ARR17* and *AGO4* were located in a series of four inverted palindromic repeats on Chr15W in *S. purpurea* [26]. In the *S. suchowensis* and *S. integra* female genomes there were only two *ARR17* copies on Chr15 instead of four, and only one *AGO4* copy instead of three. This indicates that there are only two arms of these palindromic repeats in *S. suchowensis* and *S. integra* instead of the four observed in *S. purpurea* [26]. These palindromic repeats appeared to be absent altogether in the *S. koriyanagi* and *S. viminalis* female genomes, which suggests that the palindromic repeats may have been deleted independently in *S. koriyanagi* and *S. viminalis*. Partial copies of the *ARR17* exon 1 are thought to have a key role in sex determination in both *Populus* and *Salix* subgenus *Protitea* [7,11] by silencing *ARR17* expression in males. BLAST results revealed *ARR17* exon 1 copies present on Chr15 in all *S. purpurea*, *S. suchowensis*, *S. integra*, and *S. koriyanagi* regardless of sex, while they were absent entirely from *S. viminalis* and *S. udensis*, suggesting that these partial repeats were likely lost in a common ancestor of these two species (Table 3, Appendix A). The copy number variation of *DRB1* and the three hypothetical proteins across the genomes is inconsistent with the current model of sex determination and does not support a role of these genes in sex determination, as previously proposed for *S. purpurea* [13]. Of particular interest is the lack of *ARR17* or *GATA15* homologs on Chr15 in the *S. koriyanagi* and *S. viminalis* female genomes. The missing *ARR17* in *S. viminalis* is inconsistent with earlier studies on *S. viminalis* by Hallingback et al. [30] and Almeida et al. [24], which both identified one copy of *ARR17* on the *S. viminalis* Chr15W. Taken together, the differing number of candidate sex determination genes between species, particularly *ARR17* and *GATA15*, indicates that the mechanism of sex determination may be quite labile within the *Vetrix* lineage of willows, despite its apparent conservation between other willow subgenera and the poplars.

### 3.3. Secondary Metabolism Genes

Across most genomes, the copy number of annotated secondary metabolism genes shows little variation, with a few notable exceptions. *S. suchowensis* P294 exhibited an exceptionally high copy number of several gene families, including flavonol synthase 1, terpene synthase 21 (involved in sesquiterpene synthesis), coniferyl aldehyde 5-hydroxylase (associated with kaempferol-3-O-glucoside and prunin variation [19]), and UDP-glucose flavonoid 3-O-glucosyltransferase (Appendix A). This abundance of gene annotations in P294 warrants further investigation into this particular genotype and its progeny for secondary metabolite abundance and its relationship to pollinator and pest attraction. Some other notable copy number variations between genomes included nine chalcone synthase genes in *S. viminalis* 07-MBG-5027, two copies of phytoene desaturase 1 in all three *S. suchowensis*, 20 copies of squalene monooxygenase in *S. suchowensis* P63, and 31 copies of UDP-glucose flavonoid 3-O-glucosyltransferase in *S. purpurea* 94001 [19] (Appendix A).

FPKM normalized expression results from all eight tissue types mapped to the *S. purpurea* 94006 v5.1 reference showed substantial variation in expression for secondary metabolite gene families (Appendix A). Sapur.019G055800, a 4-coumarate:CoA ligase, has been associated with phenolic glucoside production in *S. purpurea* [19]. However, both *S. purpurea* genomes had the lowest relative expression of this gene, while expression was nearly five-fold greater in both *S. koriyanagi* genotypes. *S. koriyanagi* 04-FF-016 also showed exceptionally high expression of the arogenate/prephenate dehydratase gene family, which has been associated with prunin and isosalicin production in *S. purpurea* [19]. *Salix suchowensis* P63 exhibited the greatest expression of terpene synthase 03 family genes, which are associated with numerous terpenoids including beta-ocimene, beta-pinene, farnesene, and isoprene, while *S. suchowensis* P294 exhibited the greatest expression of coniferyl aldehyde 5-hydroxylase genes associated with prunin and kaempferol-3-O-glucoside [19]. These findings suggest that further research is warranted into these genotypes to understand differences in secondary metabolite concentrations and the effects they may have on pollinator and pest attraction.

### 3.4. P336 Crosses and Progeny

Across all eight crosses generated with *S. integra* P336 as a parent or grandparent, 100% of the progeny were female. Notably, when a (*S. integra* P336 *× S. suchowensis* P63) F_1_ female was both backcrossed to *S. suchowensis* P63 and out-crossed with *S. purpurea* 94001, all of the progeny were again female. This is interesting as it suggests that all-female inheritance persists across multiple generations, despite independent assortment and recombination of autosomes. The most likely cause of such a sex bias persisting after more than one generation is the cytoplasmic inheritance of a “male killer” allele on either the chloroplast or mitochondrial genome from *S. integra* P336, such that only female gametes survive. Alternatively, ZZ progeny may survive, but a cytoplasmic factor may result in a female phenotype regardless of the state of the sex chromosomes. One likely candidate for such a factor for either of these two mechanisms is the *RPL10* gene, which was identified in every mitochondrial genome except *S. integra* P336 and *S. viminalis* ‘Jorr’ [31]. The absence of this gene is particularly striking, as its presence in the mitochondrial genome is broadly conserved across plant taxa, including gymnosperms and non-flowering plants [32]. *RPL10* encodes a protein that is a component of the 80S ribosome and plays a role in plant development and protein translation under UV-B stress, as well as antiviral signaling [33,34]. In Arabidopsis, *RPL10C* has also been found to be expressed exclusively in pollen grains, and *RPL10A* has impaired transmission in male gametophytes when either *RPL10B* or *RPL10C* are mutated [35]. The absence of *RPL10* from the *S. integra* P336 mitochondria and, therefore, all of its descendants, as well as this gene’s known role in plant and male gametophyte development, presents a compelling case for the absence of *RPL10* as the most likely explanation for the all-female bias observed in the progeny of *S. integra* P336. 

## 4. Materials and Methods

### 4.1. DNA Sequencing

Fresh young leaf tissue (approximately 100 mg) for all 11 *Salix* genotypes was collected and ground in liquid nitrogen using the Qiagen TissueLyser II with one 5 mm stainless steel bead. DNA extraction was performed using a modified CTAB-based protocol [36]. Briefly, the organic and aqueous phases were extracted using chloroform:isoamyl alcohol 24:1. After separation, a SPRI bead solution was used to select for reads greater than 1 kb [37]. For long read sequencing, 1 μg of DNA was used as input to Oxford Nanopore’s genomic DNA by ligation sequencing kit (SQK-LSK109) and the subsequent library was sequenced on an R.9.4.1 flow cell. Short-read sequencing of the same samples was performed on the Illumina HiSeq X Ten platform. 

### 4.2. RNA Sequencing

RNA was extracted from eight tissues (root, xylem, internode, node, young leaf, mature leaf, petiole, and young stem) for all 11 genotypes, as well as fasciated shoot tissue from 04-BN-051, following the protocol described in Zhang et al. [38]. Strand-specific RNA-Seq libraries were prepared by BGI and sequenced on the DNB-Seq platform, which generated paired-end 150 bp reads. The same RNA preps from mature leaves and roots were also sequenced on the Oxford Nanopore MinION platform, with the exception of *S. viminalis* ‘Jorr’, which failed quality control. The SQK-PCB109 PCR-based cDNA library kit was used to generate sequencing libraries for leaf and root tissue for all 11 genotypes and were sequenced on R.9.4.1 flow cells.

### 4.3. Hi-C Library Preparation

Hi-C libraries were prepared with the Phase Genomics Proximo Plant Hi-C kit (Phase Genomics, Seattle, WA, USA). Hi-C libraries were sequenced on the Illumina NovaSeq 6000 instrument, which generated paired-end 150 bp reads. The sequencing data of each Hi-C library underwent quality control with the phase genomics hic_qc.py script (https://github.com/phasegenomics/hic_qc; last accessed 15 November 2021) to ensure a sufficient number of informative Hi-C reads were present in each library. Hi-C heatmaps are shown in Appendix A. 

### 4.4. Genome Assembly

Assembly was performed with Oxford Nanopore reads using Flye 2.8.3 [39]. Illumina short reads were mapped to the assembled contigs with BWA-MEM [40]. Pilon and a custom python script were used to generate the corrected draft assembly with the Illumina data (Appendix A) [41]. Assembled contigs were scaffolded using Hi-C reads with Falcon [42] and Juicer Hi-C [43] to generate phased genome assemblies. A BUSCO search of the Eudicot core genes was performed against each assembly to assess the quality and completeness of each genome [44]. One assembly, 04-FF-016, produced two chimeric contigs, HiC scaffold_5 and HiC_scaffold_6, each spanning the entire length of several chromosomes. BLASTN analysis at the default parameter settings [45] was used to determine alignment to specific chromosomes, and each chimeric contig was manually cut at the approximate site where mapping behavior became abnormal. Resulting scaffolds were appended with a letter (e.g., a, b, c, etc.) to denote their origin from the original chimeric scaffold. 

### 4.5. Annotation

Genome annotation was performed with the LoReAn v2.5 pipeline [46], which utilized both Oxford Nanopore and Illumina RNA-Seq, along with protein models from the JGI *Populus trichocarpa* v4.1, *Populus deltoides* v2.1, and *Populus nigra × P. maximowiczii* v1.1 reference genome annotations obtained from Phytozome (https://phytozome-next.jgi.doe.gov; last accessed 21 March 2022) [27,47], followed by Augustus ab initio gene prediction [48]. BLASTN analysis, with the maximum target sequences set to one, was performed for each annotated transcript from every genome against the *S. purpurea* 94006 v5.1 annotation on Phytozome (https://phytozome-next.jgi.doe.gov, last accessed 6 July 2022) to identify homologous gene models [26,45]. Functional prediction of mRNAs in each annotation was performed using interproscan 5.52–86.0 [49]. The estimated number of missing genes from each annotation was determined by performing a BLASTN analysis (default parameter settings) of all *S. purpurea* 94006 v5.1 CDS sequences against all annotated genes for each genome and identifying those *S. purpurea* 94006 v5.1 genes without a match in each genome. Orthofinder was used to identify unique and shared genes for each assembly, and to generate a phylogeny tree from the annotated genes [50]. A BUSCO search of the Eudicot core genes was performed against the annotated mRNA sequences to estimate the completeness of each annotation [44]. 

### 4.6. Sex Determination Candidate Gene Analysis

BLASTN analysis of candidate sex determination genes was performed using the *S. purpurea* 94006 v5.1 [26] and *P. trichocarpa* v4.1 [47] CDS sequences of the candidate sex determination genes identified in Hyden et al. (2021) as the query, with each assembly as the target, using the default parameter settings [12]. Analyzed candidate sex determination genes included homologs of a type C cytokinin response regulator *ARR17*, a *GATA15* transcription factor, a truncated Argonaute 4 *AGO4*, a double stranded RNA-binding protein *DRB1*, and three hypothetical proteins [13]. 

### 4.7. Secondary Metabolism and Rust Gene Analysis

Analysis of candidate secondary metabolism genes was performed by creating a customized list of *S. purpurea* 94006 v5.1 gene models, which included candidate genes identified by Keefover-Ring et al. (2022) located in flavonoid, phenolic glucoside, and terpenoid QTL [19]. Genes with annotations in flavonoid and chalcone synthesis, terpene, sesquiterpene, squalene, and phytoene synthesis, and UDP-glucose flavonoid glucosyltransferase were also included, all of which have likely roles in terpenoid, flavonoid, and phenolic glucoside production. Results from the BLASTN analysis of annotated transcripts against the *S. purpurea* 94006 v5.1 reference were used to find the total matches in each respective genome for genes on the customized list of *S. purpurea* secondary metabolism genes. 

To analyze and compare expression of candidate genes, Illumina RNA-Seq data for each genome were mapped to the *S. purpurea* 94006 v5.1 reference using STAR 2.7.0 [51], read counts were determined using featureCounts [52], and FPKM calculated using EdgeR [53]. The sum of normalized FPKM values was calculated across all tissue types sequenced within each genotype and across all genes within each gene family. 

### 4.8. P336 Crosses and Progeny

To quantify female bias in progeny from the *S. integra* P336 genotype, F_1_ crosses and a select set of backcrosses were attempted with clones from each male genome in this study using the crossing method described by Kopp et al. [54]. In 2013, the 13X-426 cross was generated between P336 and 94001. In 2014, 05X-278-071, a female from a P336 × P63 cross, was crossed with 94001 and P63 to generate the 14X-454 and 14X-456 families, respectively. In 2020, P336 was crossed with *S. purpurea* ‘Fish Creek’ (94006 × 94001), a monoecious *S. purpurea* 94003 [12], P63, 04-FF-016, and 04-BN-051 to generate the 20X-565, 20X-564, 20X-278, 20X-567, and 20X-566 families, respectively. A cross with *S. viminalis* ‘Jorr’ was also attempted, but failed to produce viable seed, possibly as a result of the species being too divergent. Scoring for sex among the progeny was performed in April 2021. 

## 5. Conclusions

We present 11 new *Salix* genome assemblies and annotations as a novel resource for shrub willow breeding, genetics, and genomics that will enable more accurate genetics studies of these species in the future. This is the most comprehensive genome assembly and annotation effort to date in the genus *Salix* and represents closely related diploid species that can be compared to understand the evolution of sex determination mechanisms. We used these genomes to characterize copy number variation of interesting genes relating to sex determination and secondary metabolism, which could be drivers of dioecy through emergence of a new sex determination system or sexual antagonistic effects, respectively. We found that key sex determination genes are missing in *S. viminalis* and *S. koriyanagi* and hypothesize that a unique sex determination system exists in these species that differs from *Populus* and other *Salix* species, which further supports the dynamic nature of sex chromosome evolution in Salicaceae. We also characterized copy number variation and expression of sexually dimorphic secondary metabolite genes. Lastly, we demonstrated that *S. integra* P336 produces only female descendants and propose a missing *RPL10* gene from the mitochondrial genome as a candidate for this unusual inheritance. 

## Figures and Tables

**Table 1 ijms-24-02904-t001:** Assembly statistics of 11 genomes, with *S. purpurea* 94006 v5.1 and ‘Fish Creek’ v3.1 assemblies for comparison. ***** scaffold number of 04-FF-016 prior to manual cutting of chimeric scaffolds.

Genome	Species	Sex	Total Assembly Length	Number of Scaffolds	Number of Contigs	Contig N50 (KB)	Largest Contig (MB)	Mean Coverage	Assembly BUSCO Score
JGI v5.1 94006	*S. pupurea*	F	328,137,719	348	NA	NA	NA	NA	97.0%
JGI v3.1 ‘Fish Creek’	*S. purpurea*	M	312,123,941	274	NA	NA	NA	NA	97.2%
94006	*S. purpurea*	F	338,238,421	179	2675	319.30	4.75	72	95.8%
94001	*S. purpurea*	M	332,407,318	136	2696	232.30	3.67	55	95.8%
P63	*S. suchowensis*	M	369,253,841	135	2243	383.13	3.78	58	96.2%
P294	*S. suchowensis*	F	375,803,650	173	2589	325.52	2.46	57	95.8%
P295	*S. suchowensis*	F	382,054,263	135	1982	435.71	2.16	62	96.3%
P336	*S. integra*	F	312,752,820	111	1246	804.25	5.99	60	96.7%
SH3	*S. koriyanagi*	F	339,158,221	147	2922	335.52	2.19	45	95.5%
04-FF-016	*S. koriyanagi*	M	349,107,755	152 *	2983	300.36	2.27	75	95.1%
07-MBG-5027	*S. viminalis*	F	293,303,539	171	1716	532.84	4.16	103	95.7%
‘Jorr’	*S. viminalis*	M	282,587,186	197	2136	442.89	3.81	51	96.1%
04-BN-051	*S. udensis*	M	315,877,065	140	2087	396.09	4.45	51	95.5%

**Table 2 ijms-24-02904-t002:** Summary statistics from 11 genome annotations, with *S. purpurea* 94006 v5.1 and ‘Fish Creek’ v3.1 assemblies for comparison.

Genome	Species	Annotation BUSCO Score	Genes	Transcripts	Genes Missing	Genome-Specific Orthogroups	Genes in Specific Orthogroups
JGI v5.1 94006	*S. pupurea*	97.0%	35,125	57,462	NA	NA	NA
JGI v3.1 ‘Fish Creek’	*S. purpurea*	97.2%	34,464	46,943	NA	NA	NA
94006	*S. purpurea*	82.2%	31,938	36,199	3706	379	1026
94001	*S. purpurea*	91.1%	31,470	39,196	4164	336	770
P63	*S. suchowensis*	84.9%	30,530	37,310	4663	229	534
P294	*S. suchowensis*	89.7%	34,681	38,788	4002	298	730
P295	*S. suchowensis*	87.2%	30,719	36,507	4532	217	574
P336	*S. integra*	77.9%	29,907	34,327	4733	225	574
SH3	*S. koriyanagi*	86.1%	30,539	36,436	4973	181	442
04-FF-016	*S. koriyanagi*	87.0%	30,478	36,226	4856	229	543
07-MBG-5027	*S. viminalis*	89.0%	31,708	37,991	3732	267	706
‘Jorr’	*S. viminalis*	92.9%	30,524	34,112	4420	138	331
04-BN-051	*S. udensis*	86.5%	30,382	36,483	4902	270	609

**Table 3 ijms-24-02904-t003:** Copy number of candidate sex determination genes across the 11 annotated genomes with *S. purpurea* 94006 v5.1 and ‘Fish Creek’ v3.1 assemblies for comparison. Homologs of Sapur.019G053300 were all located on Chr19 in each assembly, and homologs of all other genes were located on Chr15 in each assembly, unless otherwise noted. * *GATA15* was originally assembled to Chr17, but this was identified as an assembly error and the region manually moved to Chr15. ** A fourth *ARR17* was identified on a purged haplotig.

Gene ID	Function	94006 JGI (F)	‘Fish Creek’ JGI (M)	94006 (F)	94001 (M)	P295 (F)	P294 (F)	P63 (M)	P336 (F)	SH3 (F)	04-FF-016 (M)	07-MBG-5027 (F)	‘Jorr’ (M)	04-BN-051 (M)
Sapur.15WG073500	ARR17	4	0	3 **	0	2	2	0	2	0	0	0	0	0
Sapur.019G053300	ARR17	2	2	2	2	1	1	1	1	1	1	1	1	1
ARR17 15Z exon 1	ARR17	1	1	1	2	1	1	1	1	1	1	0	0	0
Sapur.15WG062800	GATA15	1	0	1 *	0	1	1	0	0	0	0	0	0	0
Sapur.15WG074400	AGO4	3	0	3	0	1	1	0	1	0	0	0	0	0
Sapur.15WG074300	DRB1	2	0	5	1	1	2	1	1	5	2	2	1	2
Sapur.15WG074900	hypothetical	1	0	1	0	1	1	0	0	0	0	0	0	0
Sapur.15WG075300	hypothetical	1	0	0	0	2	2	0	2	0	0	0	0	0
Sapur.15WG075700	hypothetical	2	0	3	1	0	0	0	1	0	9	0	0	0

**Table 4 ijms-24-02904-t004:** Summary of families generated with *S. integra* P336 as the mother and maternal grandmother and resulting scores of sex on the progeny.

Family ID	Mother	Maternal Species	Father	Paternal Species	Progeny	Percent Flowering	Percent Female
13X-426	P336	*S. integra*	94001	*S. purpurea*	284	98%	100%
20X-565	P336	*S. integra*	Fish Creek	*S. purpurea*	210	75%	100%
20X-564	P336	*S. integra*	94003	*S. purpurea*	252	77%	100%
20X-278	P336	*S. integra*	P63	*S. suchowensis*	212	98%	100%
20X-567	P336	*S. integra*	04-FF-016	*S. koriyanagi*	208	97%	100%
20X-566	P336	*S. integra*	04-BN-051	*S. udensis*	204	76%	100%
14X-454	05X-278-071	*S. integra* × *S. suchowensis*	94001	*S. purpurea*	94	88%	100%
14X-456	05X-278-071	*S. integra* × *S. suchowensis*	P63	*S. suchowensis*	166	90%	100%

## Data Availability

All raw sequencing data have been deposited at the NCBI SRA (https://www.ncbi.nlm.nih.gov/sra, accessed on 12 October 2022). Raw Illumina and nanopore DNA sequencing data can be accessed with the BioProject ID PRJNA827350. The raw Illumina RNA-Seq data can be accessed with the BioProject ID PRJNA827350. Nanopore RNA-Seq data can be accessed with the BioProject ID PRJNA888070. Genome assemblies and annotations have been deposited at the NCBI Genome Portal (https://www.ncbi.nlm.nih.gov/genome, accessed on 12 October 2022) with the BioProject IDs PRJNA890276, PRJNA892589, PRJNA892593, PRJNA892594, PRJNA892596, PRJNA892597, PRJNA892598, PRJNA892599, PRJNA892600, PRJNA892601, and PRJNA892602. Protein FASTA and information files with interproscan and *S. purpurea* 94006 v5.1 BLAST results for each annotation are available on the Willowpedia github (https://github.com/Willowpedia, accessed on 10 October 2022). Genome assemblies, annotations, and annotation information files are available on Dryad at https://doi.org/10.5061/dryad.5hqbzkh9f (Released 6 December 2022).

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
