# Peer review of "De Novo Assembly and Annotation of 11 Diverse Shrub Willow (Salix) Genomes Reveals Novel Gene Organization in Sex-Linked Regions"

_ijms, 2023, doi:10.3390/ijms24032904_

Round 1

Reviewer 1 Report

In supplementary figure S6, the workflow diagram, Is “splited” the word that you want to use?

Since there appears to be plenty of dead space in figure S6, why not add the parameters used for each tool used? Makes it more repeatable and valuable as a resource.

I like that the authors have included Supplemental table S1 in addition to table 4. These are type of results that belong in the literature for other researchers to build upon.

In suppl. figure S2, the phylogenetic tree, I feel this tree could tell the story more clearly. It appears as an afterthought, a pyproduct of Orthofinder. A figure caption or legend or at least a title of what insights we can get from this tree would be useful to the reader.

Table 5 is difficult to read in the current format. I'll make a note to the editors, as that can be cleaned up in later steps.  

Abstract is acceptable. No further changes suggested.

I am not an expert in sex determination systems so I cannot address the observations described with any great scrutiny.

The opening introduction appears to cover the current state of sex determination system in this family in sufficient depth and is acceptable.

Readability is acceptable in the introduction.

For this sequence reference, (S. purpurea female that was used for the S. purpurea 94006 v5.1 reference ge-nome (https://phytozome-next.jgi.doe.gov/info/Spurpurea_v5_1, last accessed 2 Decem-ber 2022)), is there an NCBI reference as well?

The intro provides the purpose of the paper and some justification. Acceptable.

Table 1: Readable and it makes sense. The numbers are consistent across the varieties which is the hope / expectation. The BUSCOs scores are very helpful here. The n50 column does not need the precision of 2 decimal places. Remove the decimals will clean up the table.

I am glad that the row order in tables 1 and 2 match.

Table 2 could benefit from the use of commas in the numbers like in table 1. For readability.

G50 and an O50: I don’t these are common enough metrics to be used without some sort of introductory explanation. A sentence describing these 2 metrics would be appreciated.

I am not sure that the Phylogenetic analysis is needed at all in this paper. But it doesn’t harm, so leave it as is.

Tables 4 and 5 likely are better to be in supplementary due to the number of rows. Worthwhile to have though. If there was a visual that highlights the key points of that table, it would help. But is not necessary.

 Table 6 looks good.

Author Response

Response to Reviewers

Reviewer 1:

We thank the reviewer for their thorough review and detailed comments and suggestions.

  1. In supplementary figure S6, the workflow diagram, Is “splited” the word that you want to use?

Thanks for catching this - we have edited Figure S6 to correct “splited” to “split”.

  1. Since there appears to be plenty of dead space in figure S6, why not add the parameters used for each tool used? Makes it more repeatable and valuable as a resource.
    Thanks for this suggestion - we have added information on the parameters used to Figure S6 as requested.

  1. I like that the authors have included Supplemental table S1 in addition to table 4. These are type of results that belong in the literature for other researchers to build upon.

We thank the reviewer for the positive feedback. We will keep both tables in the manuscript.

  1. In suppl. figure S2, the phylogenetic tree, I feel this tree could tell the story more clearly. It appears as an afterthought, a pyproduct of Orthofinder. A figure caption or legend or at least a title of what insights we can get from this tree would be useful to the reader.
    For Figure S2 we have added text to the caption to further explain the meaning of the figure. The key finding of Fig. S2 and the reason for including it in this paper is to show that protein models within species group together, as would be expected, and thus helps validate the annotations.

  1. Table 5 is difficult to read in the current format. I'll make a note to the editors, as that can be cleaned up in later steps.

We thank the reviewer for their attention to detail. This was the result of a formatting error and has been corrected.

  1. Abstract is acceptable. No further changes suggested.

Thanks!

  1. I am not an expert in sex determination systems so I cannot address the observations described with any great scrutiny.

Thanks for your review.

  1. The opening introduction appears to cover the current state of sex determination system in this family in sufficient depth and is acceptable.

Thanks!

  1. Readability is acceptable in the introduction.

Thanks!

  1. For this sequence reference, (S. purpurea female that was used for the S. purpurea 94006 v5.1 reference genome (https://phytozome-next.jgi.doe.gov/info/Spurpurea_v5_1, last accessed 2 December 2022)), is there an NCBI reference as well?

Regarding the reviewer’s inquiry, both existing reference genomes mentioned (S. purpurea v5.1 94006 and S. purpurea v3.1 ‘Fish Creek’) are not deposited in NCBI, as per JGI policy, Phytozome is their official and only repository.

  1. The intro provides the purpose of the paper and some justification. Acceptable.

Thanks!

  1. Table 1: Readable and it makes sense. The numbers are consistent across the varieties which is the hope / expectation. The BUSCOs scores are very helpful here. The n50 column does not need the precision of 2 decimal places. Remove the decimals will clean up the table.

We have edited table 1 to round the decimal places in the N50 column.

  1. I am glad that the row order in tables 1 and 2 match.

Thanks!

  1. Table 2 could benefit from the use of commas in the numbers like in table 1. For readability.

We have edited Table 2 to include commas consistent with Table 1 as suggested.

  1. G50 and an O50: I don’t these are common enough metrics to be used without some sort of introductory explanation. A sentence describing these 2 metrics would be appreciated.

We added clarification on the definitions of O50 and G50 in the main text as requested.

  1. I am not sure that the Phylogenetic analysis is needed at all in this paper. But it doesn’t harm, so leave it as is.

We thank the reviewer for the feedback - although phylogenetic analysis was not the main point of the paper, we will leave the phylogenetic dendrogram in the supplemental so it is available if the reader is interested.

  1. Tables 4 and 5 likely are better to be in supplementary due to the number of rows. Worthwhile to have though. If there was a visual that highlights the key points of that table, it would help. But is not necessary.

Thanks for the suggestion - Tables 4 and 5 have been moved to the supplemental material and renamed Tables S2 and S3, respectively.

  1. Table 6 looks good.

We thank the reviewer for the positive feedback. This table will remain in the main text and has been renumbered Table 4.

Reviewer 2 Report

Based on this very simplified and poor presented manuscript, I can't give any more comments.

Author Response

Response to Reviewers

Reviewer 2:

  1. Based on this very simplified and poor presented manuscript, I can't give any more comments.

Since there are no specific changes suggested and the editor has recommended major revisions, we have made specific improvements throughout the manuscript as recommended by the other reviewers and editor, and we hope these are acceptable to the reviewer.

Reviewer 3 Report

Hyden et al. presented a huge genomic dataset. They tried to answer at least three questions: Whether different willows share similar sex determination genes/mechanism? Expression pattern of secondary metabolism genes that may relevant to pollinators. The reason of female biased sex ratio? These are three topics. If they insist to put them together, they should integrate them more reasonable.

1, The authors cannot use the “section” to divide the genus Salix, they can only use subgenus or clade, because no one use section for willows like this. There are a lot of evidences showed that S. arbutifolia is a member of clade Vetrix or subgenus Pleuradenia (Wu et al., 2015, 10.1186/s12862-015-0311-7; Gulyaev et al., 2022, 10.1093/aob/mcac012; Wang et al., 2023, 10.1038/s41437-022-00586-2) not their “section Salix”. They should change them in the whole manuscript. Their “subsections” is also unacceptable.

2, continued, it is better to obtain a phylogeny tree based all available willows genomes. This will help readers to get an idea the relationship of these studied willows.

3, page 2 “in order confirm the two-gene model from S. purpurea”, One should explain the “two-gene model” here.

4, “Salix are primarily insect pollinated and as such must produce a suite of secondary metabolites to attract pollinators. One can only say willows are wind- and insect-pollinated (reviewed by Zeng et al., 2022, doi: 10.11926/jtsb.4504).

5, Table 3. Are these copies from single chromosome or whole genome, how about autosomes? Is that possible to detect the sex linked regions (boundaries) of all putative 15W and 15Z they assembled. Then we can know whether these candidates in sex linked regions or not.

6, “3.2. Sex determination genes and SDR assembly”, it is hard to understand why the authors only assembled Chr15W for females. The females include chr15Z and chr15W, if they didn’t phase the two haplotypes, it is hard to believe they got real “15W”.

7, If the GATA15 on chromosome 17 is caused by misassembling, no way to fix this? Can other researcher use your newly assemblies with confidence?

8, The authors mentioned the ARR17 and GATA15 might involved in sex determination. Is that possible the chr15Z of S. integra P336 contain ARR17?

9, It is much better if the author can add abbreviation of species name to each individual (e.g. SinP336).

10, Please give specific parameters that you used for blastn or blast analysis.

11, “A cross with ‘Jorr’ was also attempted, but failed to produce viable seed.” Why? Can you give some explanations?

12, “secondary metabolism that could be a driver of dioecy” How did you get this idea in this paper?

Author Response

Response to Reviewers

Reviewer 3:

Hyden et al. presented a huge genomic dataset. They tried to answer at least three questions: Whether different willows share similar sex determination genes/mechanism? Expression pattern of secondary metabolism genes that may relevant to pollinators. The reason of female biased sex ratio? These are three topics. If they insist to put them together, they should integrate them more reasonable.

  1. The authors cannot use the “section” to divide the genusSalix, they can only use subgenus or clade, because no one use section for willows like this. There are a lot of evidences showed that  arbutifolia is a member of clade Vetrix or subgenus Pleuradenia (Wu et al., 2015, 10.1186/s12862-015-0311-7; Gulyaev et al., 2022, 10.1093/aob/mcac012; Wang et al., 2023, 10.1038/s41437-022-00586-2) not their “section Salix”. They should change them in the whole manuscript. Their “subsections” is also unacceptable.

We thank the reviewer for their detailed feedback. We have edited the manuscript to consistently refer to the subgenus classification rather than the section/subsection classification and included the appropriate references.

  1. continued, it is better to obtain a phylogeny tree based all available willows genomes. This will help readers to get an idea the relationship of these studied willows.

This is an excellent suggestion, but phylogenetic analysis of Salix species was beyond the scope of this paper (and our expertise).  A comprehensive phylogenomic analysis of the Salicaceae has been conducted by one of our collaborators - Matt Olson at Texas Tech – and a paper describing that work is currently in review. In our paper, we do include a dendrogram generated by Orthofinder (Fig. S2), which shows the relationship of each set of genome annotations to one another.

  1. page 2 “in order confirm the two-gene model from  purpurea”, One should explain the “two-gene model” here.

Thanks for this suggestion - we added more background in the main text to explain the two-gene vs single-gene sex determination system.

  1. “Salix are primarily insect pollinated and as such must produce a suite of secondary metabolites to attract pollinators”. One can only say willows are wind- and insect-pollinated (reviewed by Zeng et al., 2022, doi: 10.11926/jtsb.4504).

We agree and have clarified that Salix spp. are both wind and insect pollinated as suggested.

  1. Table 3. Are these copies from single chromosome or whole genome, how about autosomes? Is that possible to detect the sex linked regions (boundaries) of all putative 15W and 15Z they assembled. Then we can know whether these candidates in sex linked regions or not.

We added language to clarify that these copies are all on Chr15 in the sex determination regions, or on Chr19 in the case of Sapur.019G055300 homologs. We also added language to clarify that the SDR regions in each of these genotypes have been localized to Chr15 and defined by Wilkerson et al. 2022 (DOI 10.1186/s12864-021-08254-1).

  1. “3.2. Sex determination genes and SDR assembly”, it is hard to understand why the authors only assembled Chr15W for females. The females include chr15Z and chr15W, if they didn’t phase the two haplotypes, it is hard to believe they got real “15W”.

The reviewer makes an excellent point. A decision was made early in this study to only assemble a single haplotype per genome, which was done to be consistent with the overwhelming majority of public genomes that contain only a single haplotype assembly. It should be noted that there is much inconsistency on how sex chromosomes are reported, with some reporting haploid assemblies (such as previously published S. viminalis and S. suchowensis assemblies), complete diploid assemblies (monoecious S. purpurea www.ncbi.nlm.nih.gov/data-hub/genome/GCA_027405885.1), or assemblies with both sex chromosome haplotypes (such as the existing S. purpurea v5.1 genome, which only has the SDR region on Chr15 separately phased). Moreover, the inclusion of a Chr15Z assembly does not add much new information beyond Chr15W, as the amount of Chr15Z specific sequence is minimal (Zhou et al. 2020, 10.1186/s13059-020-1952-4). We included both male and female genome assemblies for S. purpurea, S. viminalis, S. suchowensis, and S. koryianagi, so Chr15Z assemblies were produced for those ZZ males and Chr15W was produced for the ZW females of these species.  Considering the short timeline the editor has imposed to return a revised manuscript, we do not have the time or resources to reassemble and reannotate the genomes to obtain distinct Chr15Z assemblies of the ZW females.

  1. If the GATA15 on chromosome 17 is caused by misassembling, no way to fix this? Can other researcher use your newly assemblies with confidence?

This is another excellent point. We have manually moved the misassembled GATA15 and flanking region from Chr17 to the Chr15 in the 94006 assembly and adjusted the gff3 file appropriately to reflect this change. We will update our Dryad genome submission with these new files prior to publication. The assembly statistics and dotplot alignments we present in the main text and supplementary material should be sufficient to ensure the reader that these assemblies are of high quality and syntenic, so any assembly errors that may exist are minimal.

  1. The authors mentioned the ARR17 and GATA15 might involved in sex determination. Is that possible the chr15Z of  integraP336 contain ARR17?

This is a perceptive question. The existence of ARR17 on Chr15Z or a second Chr15W in P336, are hypotheses we initially considered. However, such a model would only explain all-female inheritance in the F1 generation. We also see all female inheritance in two populations generated from F1 individuals (14X-454, 14X-456, Table 4 - previously Table 6), indicating that this female bias is inherited after one generation. If an “ARR17-15Z” was responsible for all-female progeny from P336, independent assortment would mean that 50% of their progeny would inherit a normal 15Z from the mother and would therefore be male. Thus, a cytoplasmic inheritance model remains the simplest and most likely explanation for all female bias inherited across multiple generations.

  1. It is much better if the author can add abbreviation of species name to each individual (e.g. SinP336).

We thank the reviewer for the suggestion. To alleviate any confusion, rather than using abbreviations, we have edited the manuscript to include the species name when mentioning each genotype in the main text.

  1. Please give specific parameters that you used for blastn or blast analysis.

We used the default blastn command line parameters for most of our blast analyses. We have noted this in the methods portion of the main text.

  1. “A cross with ‘Jorr’ was also attempted, but failed to produce viable seed.” Why? Can you give some explanations?

We added “possibly as a result of the species being too divergent” as an explanation for the failed cross between P336 and ‘Jorr’. It is not uncommon for interspecific crosses to fail and S. viminalis is somewhat distantly related to the other species.

  1. “secondary metabolism that could be a driver of dioecy” How did you get this idea in this paper?

We added “which could be a drivers of dioecy through emergence of a new sex determination system or sexual antagonistic effects, respectively” to clarify this statement.  There is clear evidence of sex dimorphism for floral secondary compounds that may influence pollinator attraction and floral defense as described in Keefover-Ring et al. 2022 J. Exp. Bot., erac260, DOI: 10.1093/jxb/erac260.

Round 2

Reviewer 3 Report

Their respones are fine.